# INFERNO: INFERRING OBJECT-CENTRIC 3D SCENE REPRESENTATIONS WITHOUT SUPERVISION

## ABSTRACT

We propose INFERNO, a method to infer object-centric representations of visual scenes without relying on annotations. Our method learns to decompose a scene into multiple objects, with each object having a structured representation that disentangles its shape, appearance and 3D pose. To impose this structure we rely on recent advances in neural 3D rendering. Each object representation defines a localized neural radiance field that is used to generate 2D views of the scene through a differentiable rendering process. Our model is subsequently trained by minimizing a reconstruction loss between inputs and corresponding rendered scenes. We empirically show that INFERNO discovers objects in a scene without supervision. We also validate the interpretability of the learned representations by manipulating inferred scenes and showing the corresponding effect in the rendered output. Finally, we demonstrate the usefulness of our 3D object representations in a visual reasoning task using the CATER dataset.

## 1 INTRODUCTION

Inferring objects and their 3D geometry in a scene is a fundamental ability of biological visual systems (Kahneman et al., 1992; Roelfsema et al., 1998; Spelke et al., 1993). Replicating this ability in machine is a promising step towards visual reasoning valuable to several applications involving object manipulation, navigation or forecasting.

Recent works (Jiang et al., 2019; Locatello et al., 2020; Burgess et al., 2019) have shown that neural networks can learn object-centric representations from low-level perceptual features. They learn to recognize the objects in a visual scene from a singe image without relying on supervision. However, most of those approaches only consider the 2D structure of images and ignore the underlying 3D geometry of the visual scenes. On the other hand, Neural Radiance Fields (NeRFs) (Mildenhall et al., 2020) have demonstrated that differentiable renderers can be combined with gradient-based optimization to learn high-fidelity 3D scene reconstructions. NeRFs have been subsequently used to learn 3D-aware generative models, including compositional scene models (Niemeyer & Geiger, 2021).

In this work, we leverage these recent advances in object-centric representation learning (Locatello et al., 2020) and 3D modelling through implicit functions (Mildenhall et al., 2020; Niemeyer & Geiger, 2021) and propose INFERNO, a model which infers a structured representation of objects and their 3D poses from a single image. Each object is represented by latent variables characterizing its shape and appearance, together with an explicit representation of their 3D poses (translation, scale and rotation). The object representations are then decoded using implicit functions that are localized in the scene according to the objects poses and combined together to generate a 2D output view. Our model does not need supervision and instead is fitted through minimizing a reconstruction loss, akin to an auto-encoder.

Disentangling the object appearance and pose in a scene representation allows for the model to manipulate a visual scene. In particular, we demonstrate that INFERNO learns interpretable object poses, which we can modify and render to alter the pose of an object in a scene. We also validate that our approach learns meaningful representations for object discovery and visual reasoning. More specifically, we show that our approach obtains competitive performance on the CLEVR6 object discovery benchmark (Johnson et al., 2017; Greff et al., 2019) as well as for the snitch localization visual reasoning task of CATER (Girdhar & Ramanan, 2019).

In summary our contributions are the following:

- We propose a model able to infer and render 3D scene representations composed of multiple objects, each of them modeled by an implicit function and explicitly localized in the scene.
- We show that the representations learned by the model are interpretable and amenable to manipulations.
- We demonstrate that the inferred representations are useful for downstream tasks by showing competitive performance in object discovery and reasoning tasks.

## 2 METHOD

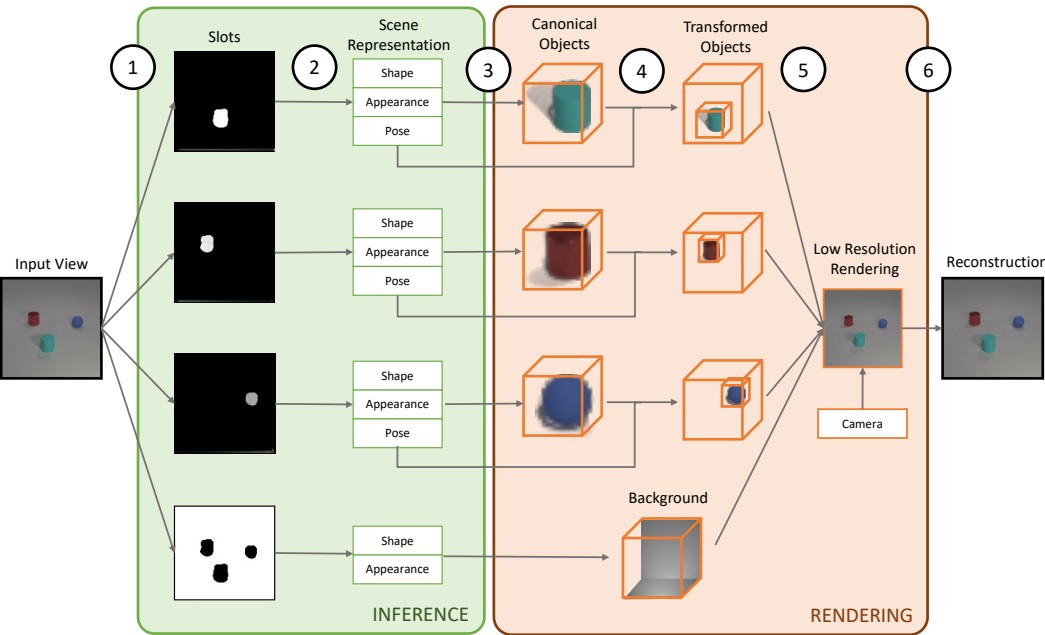

Figure 1: **Model Overview:** We propose INFERNO, a model that infers and renders object-centric 3D scene representations. **1** Our model first decomposes an input observation into multiple object slots. **2** For each slot we infer a structured 3D representation. **3** The shape and appearance determine canonical objects rendered through NeRFs. **4** Objects are transformed and located in the overall scene according to their pose. **5** We combine objects and background and render a low-resolution scene given a camera location. **6** The input is reconstructed by upscaling the low resolution scene.

We propose INFERNO (Infer NeRF Objects). The goal of our method is to infer object-centric 3D scene representations from single 2D views. Given an image $x \in \mathbb{R}^{H \times W \times 3}$, we learn an inference function $f_\theta$ that maps images to scene representations $s = f_\theta(x) = (o_1, o_2, ..., o_K, o_{bg}, c)$. Scenes are composed of $K$ objects $o_i$, a background object $o_{bg}$ and a camera location $c$.

Each object is composed of three tensors $o_i = (o_i^{shape}, o_i^{app}, o_i^{pose})$. The object shape $o_i^{shape} \in \mathbb{R}^{D_{shape}}$ and object appearance $o^{app} \in \mathbb{R}^{D_{shape}}$ are tensors that respectively describe the 3D shape occupancy and color of an object with an implicit function. The object location $o_i^{pose} \in \mathbb{R}^{4 \times 4}$ is an affine matrix that describes the object 3D pose (i.e. scale, translation and rotation) in the scene.

The background object $o^{bg} = (bg^{shape}, bg^{app})$ only models 3D shape and color, and its location is fixed, encompassing the *back-of-scene* cube. We also define a camera matrix $c \in \mathbb{R}^{3 \times 4}$, that defines the location of the scene camera and determines a 2D projection of the 3D scene.

To optimize our inference function we formulate an optimization problem in which we minimize a reconstruction loss over a dataset, similar to an auto-encoder. We define a rendering function $g_\gamma$ that takes as input a scene representation and generates a 2D view of that scene $\hat{x} = g_\gamma(f_\theta(x))$. We

assume a isotropic Gaussian likelihood model with unit covariance and optimize the probability of the data under our model, which is equivalent to minimizing the mean squared error of inputs and reconstructions:

$$\gamma^*, \theta^* = \arg\min_{\gamma,\theta} p(X|\gamma,\theta) = \arg\min_{\gamma,\theta} \frac{1}{N} \sum_{i \le N} (x_i - g_\gamma(f_\theta(x_i)))^2 \tag{1}$$

In the following sections we describe in more detail our inference mechanism, our rendering pipeline and their implementation.

## 2.1 Rendering Pipeline

We represent 3D objects as Neural Radiance Fields (NeRFs) (Mildenhall et al., 2020) with a similar setup as that of GIRAFFE (Niemeyer & Geiger, 2021). A NeRF is a function $g_\tau$ that defines a 3D shape implicitly. It takes as input a 3D location $\mathbf{l} = (x, y, z)$ and a 2D viewing direction $\mathbf{d} = (\psi, \phi)$ and outputs an occupancy value $\sigma$ and a color value $\mathbf{a} = (r, g, b)$. NeRFs are usually implemented using fully connected neural networks. Additionally, the inputs are usually embedded into a higher-dimensional space using positional encodings $\gamma$ that embed locations and viewing directions into higher dimensional spaces $\mathbb{R}^{P_l}$ and $\mathbb{R}^{P_d}$, respectively.

$$g_\tau : \mathbb{R}^{P_l} \times \mathbb{R}^{P_d} \to \mathbb{R}^+ \times \mathbb{R}^3 \quad ; \quad (\gamma(\mathbf{l}), \gamma(\mathbf{d})) \to (\sigma, \mathbf{a}) \tag{2}$$

To represent multiple 3D shapes with the same NeRF function, we can augment it with latent variables that determine which shape is being modeled (Schwarz et al., 2020). NeRFs are usually augmented with two random variables: one random variable $\mu \in \mathbb{R}^{D_{shape}}$ defines the shape of the entity being modeled, while $\upsilon \in \mathbb{R}^{D_{app}}$ models its appearance. In practice, this specialization is enforced by making the occupancy output a function of only the shape latent, while the color output is conditioned on the appearance latent.

$$g'_\tau : \mathbb{R}^{P_l} \times \mathbb{R}^{P_d} \times \mathbb{R}^{D_{shape}} \times \mathbb{R}^{D_{app}} \to \mathbb{R}^+ \times \mathbb{R}^3 \quad ; \quad (\gamma(\mathbf{l}), \gamma(\mathbf{d}), \mu, \upsilon) \to (\sigma, \mathbf{a}) \tag{3}$$

In INFERNO, we share a single parametrization of a NeRF function across all objects. Each object specific shape and appearance is defined by the shape and appearance latent variables, which correspond to the object attributes $o^{shape}, o^{app}$. The background is defined as another NeRF with separate parameters. The background NeRF also has shape and appearance latent variables to model different backgrounds.

The pose of an object $o_i$ in the scene is determined by the affine transformation matrix $o_i^{pose}$. We denote the coordinate system of the NeRF function of an object as the object space, and the coordinate system of the scene (and the background NeRF) as scene space. Given an object pose, we can convert points from the scene space to the object space by applying the $o^{pose}$ transformation matrix on those points, and we can transform points from object space to scene space by computing the inverse of the object pose matrix.

To render a scene, we cast rays from each pixel in the 2D plane defined by a given camera to the 3D scene. We evaluate NeRFs at different points along a given ray, and integrate their occupancy and color outputs to determine pixel values. Rays might traverse multiple object NeRFs in addition to the background NeRF. To determine the occupancy and color of points described by multiple NeRFs, we first query each NeRF at those particular points. To query the object NeRFs, we first need to transform the points from scene space to the particular object space. Then, we compose the results of each NeRF with a pooling function $C$, which in our case is a weighted average:

$$C(l, \mathbf{d}) = (\sigma = \sum_{i=1}^{N} \sigma_i, \frac{1}{\sigma} \sum_{i=1}^{N} \sigma_i \mathbf{a}_i) \tag{4}$$

Since rendering with NeRFs as originally proposed is computationally expensive, we render output views at a fixed low resolution. Low resolution scenes are then upscaled to the desired output

resolution using a convolutional neural network, keeping the entire rendering pipeline differentiable. Additionally, low resolution scenes are rendered with additional channels, allowing for more detailed upscalings beyond those possible when just rendering low resolution RGB outputs. For more details about NeRF rendering and compositional NeRF models refer to (Mildenhall et al., 2020; Niemeyer & Geiger, 2021).

## 2.2 INFERENCE

Given the rendering pipeline, the goal of our method is to infer representations that reconstruct a given scene. Our inference mechanism computes image features through a neural network encoder and then extracts $K$ object and a background slots. These slots are then mapped to our structured scene representation through learned neural networks.

To extract image features for each object and background slots, we use Slot Attention (Locatello et al., 2020). Slot Attention is a mechanism that maps a set of $K$ entities, called slots, to image features without annotations. It extracts image features $I \in \mathbb{R}^{H \times W \times D}$ from a given input using a resolution-preserving convolutional encoder. These features are then attended to by a set of $K$ randomly sampled slots $\pi_j \sim \mathcal{N}(\mu, \sigma)$ of dimension $D$, where $\mu$ and $\sigma$ are learnable parameters. We denote by $\pi$ the matrix concatenating all the sampled slots. Slots attend spatial chunks of the input features $I$ through soft-attention $u = TQ^T$, where $T = k(I)$ and $Q = q(\pi)$ are the embeddings of the inputs and slots respectively. The attention weights are normalized through a softmax operating on the slots axis, which makes slots compete among themselves and discourages multiple slots from attending the same input region $w = softmax(u)$. The weighted average of $I$ according to the attention weights is then computed and fed to a GRU network, to update each slot value: $\pi_j = GRU(w_j * I, \pi_j) \ \forall j \in 1, K$. Multiple rounds of soft attention are performed to iteratively refine the slots. For more details about Slot-Attention, please refer to (Locatello et al., 2020).

Object slots are unstructured tensors that result from aggregating image features. We map these slots to our structured scene representation through small fully-connected networks that operate on individual objects. More concretely, we map object slots to their 3D pose in the scene through a 2-layer MLP. To infer the object shape and appearance tensors we also use 2-layer MLPs, but we make them conditional on the predicted object pose through conditional normalization (Dumoulin et al., 2018).

## 3 RELATED WORK

### 3.1 3D SHAPE REPRESENTATIONS

There are different ways to represent 3D geometry such as voxels or meshes (Rematas & Ferrari, 2020; Gkioxari et al., 2019). For example, the GAN models of Nguyen-Phuoc et al. (2019; 2020) successfully use voxel-based representations to render images. Voxel-based methods have trouble scaling up to high resolutions as the size of a voxel representation scales cubically with the resolution.

Recently, the use of functions that implicitly model 3D volumes has gained popularity (Park et al., 2019; Mescheder et al., 2019; Sitzmann et al., 2019; 2020b;a; Kosiorek et al., 2021; Pumarola et al., 2021; Yu et al., 2021a). Implicit representations have better scaling properties, as usually the output resolution does not directly affect the dimensionality of the learned function. NeRFs (Mildenhall et al., 2020) generate scenes by learning a function that outputs the occupancy and color of points in a scene when viewed from a particular direction. By casting rays through a plane and aggregating the output values NeRFs can generate 2D views of an implicitly modeled 3D scenes. NeRFs have obtained superior reconstructions compared to other implicit methods, and our model uses NeRFs to represent multiple objects and the background of a scene.

Most methods using NeRFs represent scenes monolithically as a single entity. GIRAFFE (Niemeyer & Geiger, 2021) is a GAN-based method that represents multiple objects in a scene using NeRFs as part of their generator. Their factored representations are amenable to object manipulations. Our model uses a rendering pipeline inspired by GIRAFFE. However, we focus on recovering scene representations from existing images, while GIRAFFE does not have an inference mechanism. Ob-jSURF (Stelzner et al., 2021) and UORF (Yu et al., 2021b) infer scene representations composed

of multiple objects, each object represented with a differently instantiated NeRF. Different from our work, they focus on novel view generation. These methods do not explicitly infer the pose of the different objects in the scene. Both methods use ground-truth camera locations and multiple scene views during training, and ObjSURF additionally requires depth annotations.

## 3.2 OBJECT-CENTRIC SCENE MODELS

Segmenting objects in a scene is a landmark computer vision task with an extensive literature. Recently, there is a line of work on object-centric generative models of scenes (Kosiorek et al., 2021; Burgess et al., 2019; Locatello et al., 2020; Lin et al., 2020; Greff et al., 2019). These models learn to generate scenes as a composition of multiple objects and a background. When equipped with an inference mechanism, these models learn to segment objects in a scene without annotations, driven by their compositional generative process. MONet (Burgess et al., 2019) implements a multi-object VAE that segments object sequentially by infering latents corresponding over parts of the scene not yet attended to iteratively. IODINE (Greff et al., 2019) uses a similar multi-object VAE and performs multiple rounds of inference to settle on a scene decomposition. Slot-Attention (Locatello et al., 2020) maps a set of entities, called slots, to image features through multiple rounds of soft attention. The slots compete among themselves to attend to features, making each slot attend to a region of the input image. When driven with alpha-compositing decoder, slot attention learns to segment objects in a scene. Our model uses a variant of Slot Attention to decide on which part of a 2D view should each object in our scene attend to, but we infer 3D-aware representations for each object. Object-centric scene models have also been implemented as world models, with the goal of simulating dynamics (Lin et al., 2020). By decomposing the scene into objects, these methods can simulate dynamics at an object level, which are usually simpler and shared among objects. Contrary to most previous approaches which segment 2D shapes, our method infers object-centric scene representations in 3D space.

## 4 EXPERIMENTS

In this section we showcase the capabilities of INFERNO with three main experiments. First, we demonstrate the interpretability of the scene representations through manipulating scenes and verifying the corresponding effects in the rendered outputs. Then we show that it learns to identify and segment the objects in a scene without supervision. Finally, we highlight the usefulness of such representations for downstream tasks by applying our model on the CATER snitch localization task.

### 4.1 TRAINING SETUP

We use the same training setup for all experiments unless otherwise mentioned. We train our models for 400K iterations using the Adam optimizer Kingma & Ba (2014) with a learning rate of $1 \times 10^{-4}$. We use a batch size of 128 and we use up to 16 nVidia V100 GPUs. We use learning rate warmup (Goyal et al., 2017), which was found to be helpful to avoid optimization issues with Slot Attention. We use a weight decay rate $\lambda = 1 \times 10^{-6}$. We use three iterations of slot attention during training and evaluation. We set up the number of objects in a scene as the maximum possible number of objects in a dataset, i.e. five objects for CLEVR2345, 6 for CLEVR6 and 10 for CATER. Please refer to Appendix B for more details on the experimental setting.

### 4.2 SCENE INFERENCE

In this section we demonstrate the properties of our scene representation. Our model infers 3D object-centric scene representations from single 2D views. These representations disentangle the 3D appearance and pose of objects, which allows for semantic manipulations of the scene not possible otherwise. These manipulations can be validated by rendering the modified scene representations. Additionally, we verify that decomposing the scene into multiple objects leads to better reconstructions.

First, we verify the quality of INFERNO's generations by comparing them two baselines: i) a version of our model that does not consider multiple objects, and ii) the GAN method of GI-RAFFE (Niemeyer & Geiger, 2021). We perform this comparison on the CLEVR-2345 dataset

| Input | Recon. | Addition | Removal | Scale | Forward | Right |

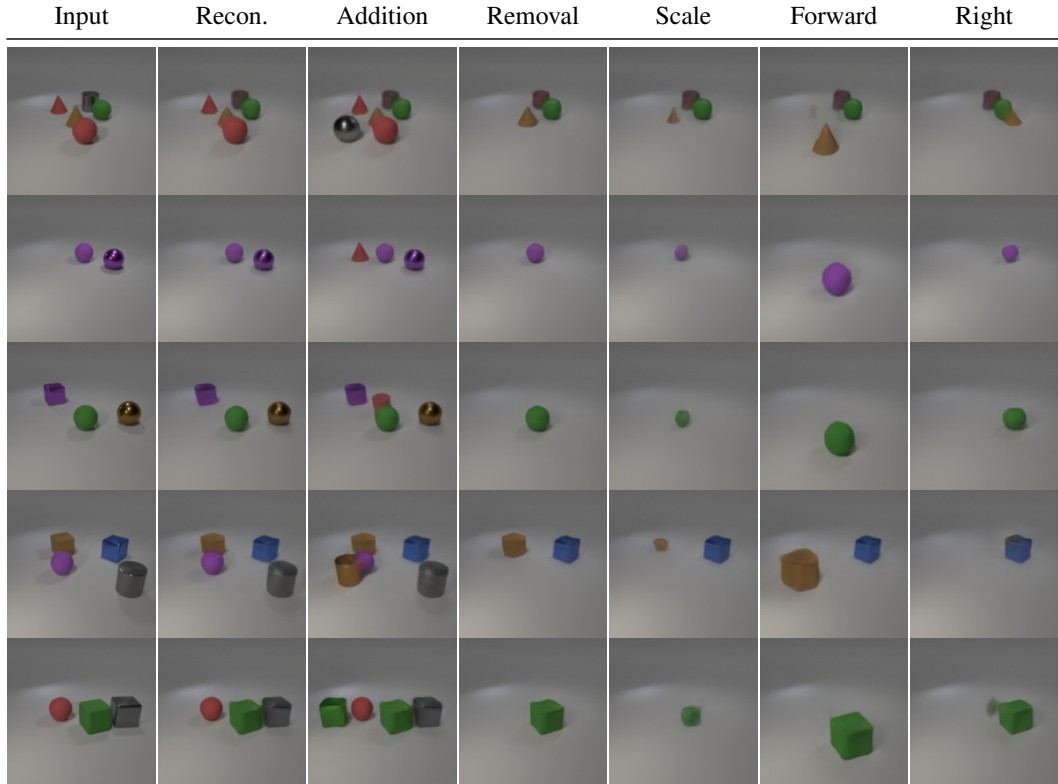

Figure 2: **Manipulations on CLEVR**: we show some examples of the manipulations we perform to CLEVR2345 images, including object removal and addition, changing the scale of an object, and object translation. Our model can perform these transformations because it disentangles object pose and appearance.

Table 1: **Reconstruction error on CLEVR2345** We consider an autoencoder baseline that uses a single NeRF object covering the whole scene (NeRF-AE), and compare it to our model on the test set of CLEVR-2345. NeRF-AE struggles to reconstruct multiple objects accurately. In contrast, INFERNO obtains much more precise reconstructions under all metrics and allows for object identity/pose manipulations.

| Model | MSE ($\downarrow$) | PSNR ($\uparrow$) | SSIM ($\uparrow$) | LPIPS ($\downarrow$) |
|---|---|---|---|---|
| NeRF-AE | $5.14 \times 10^{-4}$ | 44.89 | 59.24 % | $168.9 \times 10^{-3}$ |
| Ours | $1.22 \times 10^{-4}$ | 52.07 | 72.4 % | $18.93 \times 10^{-3}$ |

introduced by GIRAFFE, which contains CLEVR images with 2 to 5 objects. For reconstruction, we compare models using reconstruction metrics including mean-squared error, PSNR and SSIM. We rely on the population metric Frechet Inception Distance (FID) to evaluate the generation quality.

In INFERNO, we generate novel scenes by inferring representations for ground-truth images and then manipulating them. To compare to GIRAFFE, we manipulate scenes by adding additional objects and swapping object shapes and appearances across scenes. Manipulations are described in more detail in the Appendix. We highlight that GIRAFFE is an unconditional model, while INFERNO generates novel scenes conditioned on existing ones.

We also investigate different interpretable manipulations of scene representations and visualize the effects in the corresponding output renderings. We also conduct this experiment on the CLEVR-2345 dataset. We validate that our model is able to render out-of-distribution scenes not corresponding to training examples, such as scenes having 1 or 6 objects, and verify that the pose manipulations have semantically coherent effects.

Table 2: **FID on CLEVR2345** We consider our model as a NeRF scene generator and compare it to the state-of-the-art. When reconstructing ground-truth images, our model obtains better FID than GIRAFFE. We then consider object manipulations to generate novel scenes from existing ones. While adding new objects to a scene slightly increases our FID score, when exchanging object identities across scenes we set a new state-of-the-art for image generation on CLEVR2345.

| Model | FID ($\downarrow$) |
|---|---|
| GIRAFFE | 37.7 |
| Ours - Reconstruction | 23.5 |
| Ours - Add Object | 27.2 |
| Ours - Swap Object | 23.7 |

Table 1 shows the reconstruction metrics obtained by our model and baseline on the CLEVR2345 dataset. Note that GIRAFFE is a GAN-method that does not have an inference mechanism, and therefore it cannot reconstruct scenes. We observe that the NeRF-AE baseline obtains higher reconstruction error than our regular model, as our object-centric method can make better use of its capacity. In Table 2 we compare INFERNO with GIRAFFE using the FID metric. Our model reconstructions have better FID than the generations of the GIRAFFE. Additionally, our model can perform inference and manipulations on existing scenes. We use that capability to generate novel scenes by manipulating existing ones. Our model is able to generate novel scenes with additional objects or with altered object shapes and appearances, with better FID than GIRAFFE.

In Figure 2 we show some examples of the scene manipulations possible with our model. Given a scene representation, we can remove or add objects, rearrange object poses, translate the objects to new locations or change the object scales. While some of these manipulations can be performed with regular object-centric models, modifications to the scale and location of the objects are hard to implement without explicitly modeling 3D object pose.

## 4.3 OBJECT DISCOVERY

Table 3: **Object Discovery on CLEVR6** INFERNO, despite inferring more complex 3D object segmentations without annotations, is competitive with the current state-of-the-art 2D object discovery methods on CLEVR6.

| Model | ARI % ($\uparrow$) |
|---|---|
| Slot-Attention | $98.8 \pm 0.3$ |
| IODINE | $98.8 \pm 0.0$ |
| MONet | $96.2 \pm 0.6$ |
| Slot MLP | $60.4 \pm 6.6$ |
| Ours | $96.7 \pm 0.2$ |

Unsupervised object discovery consists in learning to segment the objects in a scene without using annotations. We test our model on CLEVR6 benchmark, a variant of the CLEVR dataset with scenes of up to 6 objects and annotated with 2D object masks. We choose this dataset to compare to previous work in unsupervised object discovery. Note that this setup evaluates 2D segmentation masks, although our model naturally provides 3D segmentations. We evaluate the quality of the segmentations using the Adjusted Random Index (ARI) metric (Rand, 1971), which is a measure of clustering similarity. In line with previous work, we compute only the foreground ARI, which does not take into account the background segmentation mask. In particular, we consider object as a different clusters and compare the cluster assignment of each foreground pixel in the original image to its prediction. To determine which pixels correspond to each object in our model we make use of the input segmentation masks predicted by our inference mechanism.

Table 3 reports the ARI metric of our model and different baselines in this task. We observe that IN-FERNO is competitive with state-of-the-art methods, surpassing MONet and having slightly lower

| Input | Background | Object 1 | Object 2 | Object 3 | Object 4 | Object 5 | Object 6 |
|-------|------------|----------|----------|----------|----------|----------|----------|

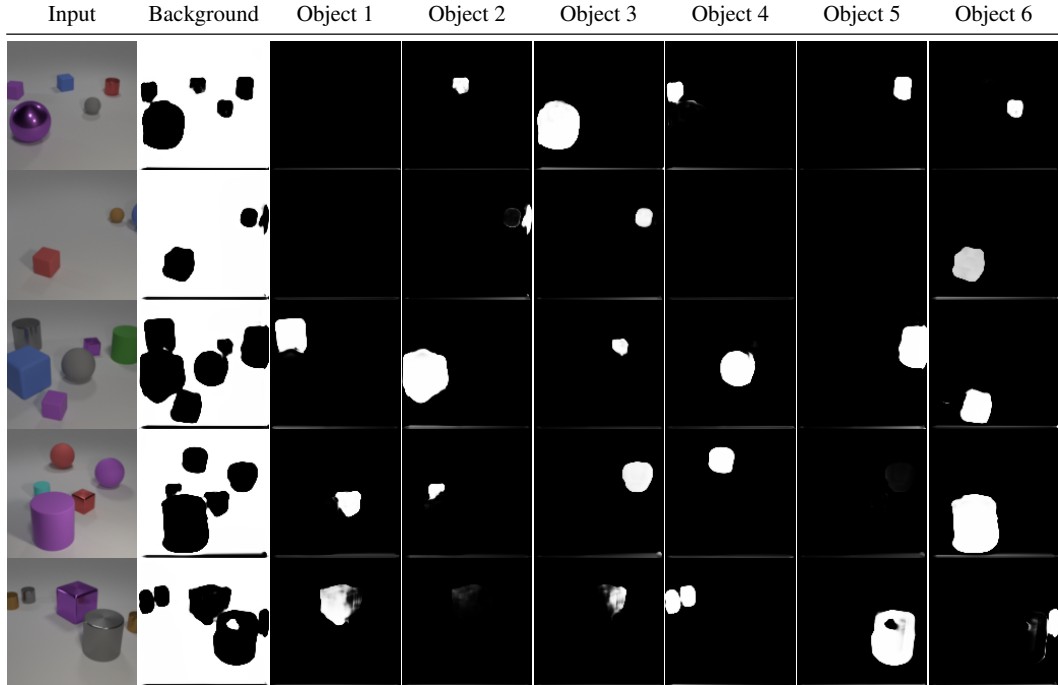

Figure 3: **Object Discovery on CLEVR6**: INFERNO identifies the different objects in a scene without supervision. For each input image, we show which regions of the input are attended by each object as well as the background. We include an example of a failed segmentation in the last row, where one object slot (4) is trying to represent multiple objects at the same time.

ARI than Slot-Attention and IODINE. However, our model learns to segment objects in 3D, while the other baselines extract 2D segmentation masks. Figure 3 shows some examples of the discovered object masks. We can see that the model learns to segment different objects and properly discards object slots when the number of objects in the scene is lower than needed. We also include an example that our model fails to segment properly - multiple objects are segmented by the same object slot, and a single object is represented in multiple parts by different slots.

## 4.4 SNITCH LOCALIZATION

The goal of this experiment is to show that the representation learned by our model is useful for the snitch localization task from the CATER dataset (Girdhar & Ramanan, 2019). We focus on the CATER task involving videos with a static camera. The objective is to predict the final position of an object (the snitch) in a video. The scenes show multiple objects that move over time, one being the snitch object. The snitch can be occluded and moved around simultaneously by other objects, requiring object tracking and reasoning about dynamics to solve the task.

We follow the experimental setup of Ding et al. (2020). First, we train INFERNO to reconstruct images from the CATER dataset. Once INFERNO is trained, we discard the rendering pipeline of our model, and instead feed the scene representations to a 12-layer transformer model to predict the final snitch position. Each object in our representation is given as an input element to the transformer. We add a learned positional encoding to the object representations based on their frame index. Objects in the same frame have the same positional encoding. The transformer last output is given to a MLP head that predicts the logits for the 36 possible output positions.

We minimize the sum of the cross-entropy and a L1 loss between the predicted and the true snitch final position. Following (Ding et al., 2020), we also consider the use of auxiliary SSL loss after pretraining. The SSL loss randomly masks one object per-frame and tries to predict its representation at the corresponding output. The model minimizes the L2 distance between the predicted object

Table 4: **Snitch Localization on CATER**. We report Top-1 and Top-5 accuracies for the snitch localization task. Our model outperforms the R3D LSTM and R3D NL LSTM models that learn unstructured representation. It indicates that the structured representation learned by INFERNO is useful for this task. INFERNO pretraining is also critical, showing that the pretraining and not the encoder architecture is a key component. Overall, INFERNO achieves performances close to the state-of-art approaches.

| Model | Top-1 | Top-5 |
|---|---|---|
| R3D LSTM | 60.2 | 81.8 |
| R3D + NL LSTM | 46.2 | 69.9 |
| OPNet (extra anns.) | 74.8 | - |
| Hopper | 73.2 | 93.8 |
| Slot-Attention | 59.1 | 88.0 |
| Aloe (w/out SSL loss) | 60.1 | - |
| Aloe | 74.0 | 94.0 |
| Ours (w/out pretraining) | 2.91 | 12.9 |
| Ours (w/out SSL loss) | 69.17 | 87.68 |
| Ours | 71.7 | 88.9 |

representation and the observed one. The SSL loss is only backpropagated through the transformer and not the inference network.

During training, we randomly samples 40 frames from a video and predict the snitch localization from this one crop. At test time, we randomly sample 10 temporal crops of 40 frames each and average the prediction over the 10 crops as our final prediction. Refer to Appendix B for more details about the experiment setup.

Table 4 reports CATER Top-1 and Top-5 accuracies for different methods. We first compare our model that is pretrained to reconstruct image on CATER with a randomly initialized encoder. Using a randomly initialized encoder does not perform well. It indicates that INFERNO pretraining, and not the encoder architecture, is key to learn useful representation for the task. We also observe that the additional SSL loss provides regularization and slightly improves the performance of our model.

We next compare our approach with Aloe (Ding et al., 2020), an object-centric baseline, R3D and R3D NL, 3D convolutional models proposed by (Girdhar & Ramanan, 2019) and architectures using strong inductive biases toward object tracking such as OPNet (Shamsian et al., 2020) or Hopper (Zhou et al., 2021). Our model outperforms the R3D and R3D NL models that rely on unstructured representation. This result suggests that the object-centric representation learned by INFERNO is useful for the visual reasoning task. Our model also outperforms Aloe, an object-centric method using 2D object representation, when both methods do not use additional SSL loss. However Aloe benefits more from the use of an additional SSL loss. Overall, INFERNO achieves performances close to the state-of-art approaches.

We finally evaluate the performance of a slot-attention baseline (Locatello et al., 2020) in Table 4. The slot-attention baseline first pretrains a slot-attention encoder, with a similar architecture than our model, by reconstructing CATER frames using a mask decoder. It then fine-tunes the encoder using the same procedure than our model to solve the snitch localization task. We observe that our model significantly outperforms the slot-attention model which focus on the 2D geometry of the scene. This result supports the advantage of 3D-aware representation for solving the CATER task.

## 5 CONCLUSIONS

We propose INFERNO, a model for inferring object-centric 3D scene representations. Our model is able to discover objects in a scene without annotations, and the inferred scene representations are interpretable and amenable to manipulations. Further, the scene representation is useful for visual reasoning downstream tasks such as the snitch localization task in CATER.

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

## A ADDITIONAL MODEL DETAILS

Our model is composed of five main modules: encoder, slot attention, slot to object mapping, NeRF decoder, neural network upscaler. In this section we provide additional details about each of these modules as well as describe their architecture.

Table 5: **Encoder Neural Network**

| Layer Type | Size | Normalization | Activation | Other details |
|---|---|---|---|---|
| Conv $5 \times 5$ | 64 | - | ReLU | Stride 1 Pad. 1 |
| Conv $5 \times 5$ | 64 | - | ReLU | Stride 1 Pad. 1 |
| Conv $5 \times 5$ | 64 | - | ReLU | Stride 1 Pad. 1 |
| Conv $5 \times 5$ | 64 | - | ReLU | Stride 1 Pad. 1 |

**Encoder** The goal of the encoder is to extract image features. We use an encoder with no downsampling, as it is typically used with Slot Attention. Details about the encoder architecture are described in Table 5.

Table 6: **Slot Attention Neural Network**

| Name | Size | Description |
|---|---|---|
| Positional emb. | 64 | Additive embedding, same size as CNN input features |
| Flatten | - | Flattens the spatial dimensions of CNN features |
| QKV MLP | 128 | Linear layers that map slots and input features to the same dimension |
| LayerNorm | 128 | Normalizes the slots/inputs |
| MLP + GRU | 128 | The output of soft-attention goes through a linear layer + GRU |

**Slot Attention** We employ Slot Attention to map image features to object slots. Slots are sampled randomly from a Gaussian distribution with learned parameters. We use different distributions for the background slot and the object slots. During training we employ three iterations of slot attention to refine the image features to slot assignments.

Table 7: **Slot to Object MLP details**

| MLP Name | Size | Act and Norm. | Description |
|---|---|---|---|
| Obj Pose | 7 | ReLU, LayerNorm | Slot to translation, scale and rotation |
| Obj Shape/App. | 128 | ReLU, CondLayerNorm | Slot to shape and appearance |
| BG Shape/App. | 128 | ReLU, LayerNorm | Background slot to its shape/app, fixed pose. |

**Slot to Object Net** The background and object slots are mapped to scene parameters using a series of MLP. For each object slot, we first map the object to its pose parameters. We use a 2-layer MLP with LayerNorm to map a slot to its pose. The size of the hidden dimension is the same than

the output dimension size. We parametrize object pose as a 7-dimensional tensor. We use three dimensions for the object location along each axis, three dimensions for the scale of the object and a single dimension to express a rotation of the object along the X axis. These parameters are then mapped to their corresponding $4 \times 4$ affine transformation matrix for each object. Note that in practice we are not modeling rotations in the experimental section.

Once we have inferred the object poses, we infer object shapes and appearances. These are inferred individually for each object using a common 2-layer MLP. The MLPs are conditional on the object pose using Conditional Layer Normalization, that makes the learned parameters of LayerNorm be a function of a condition. Specifically, we map the 7-dimensional pose tensor to LayerNorm parameters with a single linear layer with no activation or normalization. Shapes and appearances are defined by the output of the MLPs, which produce two 128-dimensional tensors.

For the background object we only infer shape and appearance, and define its pose to be that of the scene cube. The shape and appearance of the background are inferred through another 2-layer MLP with ReLU and LayerNorm.

Note that our scene representation also admits a camera pose. In our experiments we fix the camera location to look at the scene from a standard location (centered and 33 degrees above the Z plane). Other concurrent approaches (Yu et al., 2021b; Stelzner et al., 2021) use ground-truth camera locations to generate novel views, while we focus on recovering scene representations without the use of ground-truth annotations.

Table 8: **Details about the NeRF MLPs used**

| MLP Name | Layers | Size | Description |
|----------|--------|------|-------------|
| Obj MLP | 8 | 64 | ReLU activation, no norm. Skip connection with layer 4. |
| BG MLP | 4 | 16 | ReLU activation , no norm. |

**NeRF MLPs** With the object shape and appearance tensors we can render them following GRAF (Schwarz et al., 2020). We use one NeRF for the objects and one NeRF MLP for the background. The details about each NeRF architecture can be found in Table 8. Note that, to render objects according to their pose, we query their NeRF MLP in a canonical object space by transforming input coordinates in scene space to object space using the object pose. To reduce the computational complexity of rendering with many NeRFs, we render scenes at a fixed resolution of 16x16. Instead of rendering RGB pixels, we render feature images with 128 channels. The output of the rendering is then upscaled and mapped to RGB views with a neural network upscaler.

Table 9: **Neural Upscaler architecture**

| Layer | Size | Activation | Normalization | Other |
|-------|------|------------|---------------|-------|
| Conv $3 \times 3$ | 64 | ReLU | Instance | Stride=1 Pad=1 |
| Upsample | - | - | - | Nearest Neighbors |
| Conv $3 \times 3$ | 64 | ReLU | Instance | Stride=1 Pad=1 |
| Upsample | - | - | - | Nearest Neighbors |
| (Only 128px) Conv $3 \times 3$ | 64 | ReLU | Instance | Stride=1 Pad=1 |
| (Only 128px) Upsample | - | - | - | Nearest Neighbors |
| Conv $3 \times 3$ | 3 | - | - | Stride=1 Pad=1 |

**Neural Upscaler** The neural upscaler takes the low resolution output of the NeRF MLPs and upscales it to the full output resolution. Additionally, it maps the rendered image to RGB space. This module is implemented using a convolutional neural network. We always render the NeRF output at 16px. Consequently, we add additional layers to the neural upscaler depending on the desired output resolution. For most experiments we use a resolution of 64px, while for the Scene Inference experiments on CLEVR2345 we use a resolution of 128px. The architecture of the neural upscaler can be found in Table 9.

# B EXPERIMENT DETAILS

## B.1 SCENE MANIPULATION

**Dataset**  For generating manipulated scenes we consider the CLEVR2345 dataset (Niemeyer & Geiger, 2021). Images in the CLEVR2345 dataset contain from 2 to 5 objects. We use the original train and test splits. Images are resized to 128x128 pixels and RGB values are normalized in the [0, 1] range.

**Training**  We use a batch size of 128 and train our model for 400k iterations. We use Adam with a learning rate of $1 \times 10^{-4}$ and weight decay $1 \times 10^{-6}$. We use 5 objects and rely on the model to not use additional slots if the scene shows less than 5 objects. We use the neural upscaler with additional layers to upscale to 128px. Additionally, for this experiment we use an additional LPIPS loss. We use the LPIPS metric computed by an AlexNet network, and we add this loss to our regular MSE loss. We weight the LPIPS loss by a factor of 100, so that it has a comparable order of magnitude to the MSE loss.

**Manipulations**  Manipulations are done as follows:

- *Substraction:* We randomly delete up to two of the object slots.

- *Addition:* We randomly add an object slot from another scene.

- *Scale:* We reduce the scale (in all XYZ axis) of one of the objects in the substraction scene.

- *Forward:* We manipulate the pose vector of one object in the substraction scene and move it forward on the Z axis.

- *Right:* We manipulate the pose vector of one object in the substraction scene and move it forward on the X axis.

Additionally, we consider the *Swap* transformation for Table 1. This transformations modifies a scene by replacing the object shape and appearance vectors with those of an object from another scene.

**Metrics**  To compare reconstructions we use Mean-Squared Error, Peak Signal-to-Noise Ratio (PSNR), Structural Similarity (SSIM) and the LPIPS metrics.

MSE measure the average squared difference between pixel values.

$$\text{MSE}(x, x') = \frac{1}{N} \sum_N (x - x')^2 \tag{5}$$

PSNR is a metric commonly used in signal processing.

$$\text{PSNR}(x, x') = -10 \log_{10}(\text{MSE}(x, x')) \tag{6}$$

SSIM (Wang et al., 2004) provides scores more aligned with human perception, specially under the presence of image noise. Scores are computed convolutionally by applying a kernel over images, which are then contrasted.

LPIPS (Zhang et al., 2018) computes differences in neural network activations for two images. It is a perceptual metric that has been shown to have higher correlation to human perception than other metrics not based on neural networks.

To compare populations of generated images we use the Frechet Inception Distance (Heusel et al., 2017). The Frechet Inception Distance embeds images into a neural network space and then fits a Gaussian distribution to the generated and ground-truth activation statistics. The score is obtained by then computing the Frechet distance between the two. Note that other metrics such as Inception Score are not applicable for the CLEVR2345 since there are no well-defined classes.

## B.2 OBJECT DISCOVERY

**Dataset** For object discovery we consider the CLEVR6 dataset. We use the original CLEVR6 dataset and extract the images from TFRecord files available at this URL. We use the original training/test split, using the first 70% images for training and the remaining ones for test. We take a crop between pixels [29, 221] and [64, 256], for the height and width respectively, and then resize the crop to 64px. We normalize the value of the images between [0, 1]. To generate the CLEVR6 dataset, we keep only those images that have at maximum 6 objects according to the annotation files.

**Training** We use a batch size of 128 and train our model for 400k iterations. We use Adam with a learning rate of $1 \times 10^{-4}$ and weight decay $1 \times 10^{-6}$. We use 6 objects and rely on the model to not use additional slots when needed.

**Metrics** We follow previous work (Greff et al., 2019) and use the Adjusted Rand Index (ARI) (Rand, 1971) to evaluate cluster assignments in object discovery. ARI scores range from 0 (random assigment) to 1 (perfect match). As in previous works, we do not consider a segmentation mask for the background.

## B.3 VISUAL REASONING ON CATER

For the visual reasoning task, we consider the CATER dataset and uses 5K videos that do not have camera motion. All the videos are resized to a 64x64 resolution.

### B.3.1 PRETRAINING

We first pretrain our INFERNO to reconstruct individual frames from the CATER dataset. We bootstrap a model trained on CLEVR6 for object discovery for 400k iterations and train it on the CATER dataset for an additional 100k iterations to speed-up training, as the iteration time of a model with 10 objects is larger. We use a batch size of 128 and we use Adam with a learning rate of $1 \times 10^{-4}$ and weight decay $1 \times 10^{-6}$. When training on CLEVR6 we use 6 object slots, while when training on CATER we use 10 object slots.

### B.3.2 FINETUNING

After pretraining, we finetune the INFERNO encoder to the supervised task of snitch localization. We discard the rendering pipeline of our model, and instead feed the inferred object slots representations to a transformer that aims at predicting the final position of the snitch.

To predict the snitch, we consider a 12 layers transformer with the hidden dimension of 128 which takes the slot representation as input. The transformer treats each object as input element. A learned positional embedding is added each slots representation based on their frames index, i.e. the position of the objects is the same within a frame. The final output to the transformer is given to a 1 layer MLP head with an hidden dimension of 128. It outputs 36 logits that correspond to possible snitch location. We minimize the sum of the cross-entropy between the predicted position and the true target and a the l1 loss between the prediction and target.

We optionally use a SSL loss similar to Ding et al. (2020). The SSL loss randomly masks one object per-frame and tries to predict its representation at the corresponding output. The model minimizes the L2 distance between the predicted object representation and the observed one. The SSL loss is only backpropagated through the transformer and not the encoder. We weight the SSL loss by a factor of $1.0e - 3$.

We use an Adam optimizer to minimize the loss. The initial learning rate is set to the $1.0e - 4$ and gradually decreased to $1.0e - 6$ using a cosine learning rate decay. Similarly, we use a initial weight decay of $1.0e - 5$ that we increases to $1.0e - 3$ using a cosine schedule. Our model is finetuned for 500 epochs. We don't make use of learning rate decay.

During training, we randomly samples 40 frames from a video and predict the snitch localization from this one crop. At test time, we randomly sample 10 temporal crop of 40 frames each and average the prediction over the 10 crops as our final prediction.

## C  ADDITIONAL VISUALIZATIONS

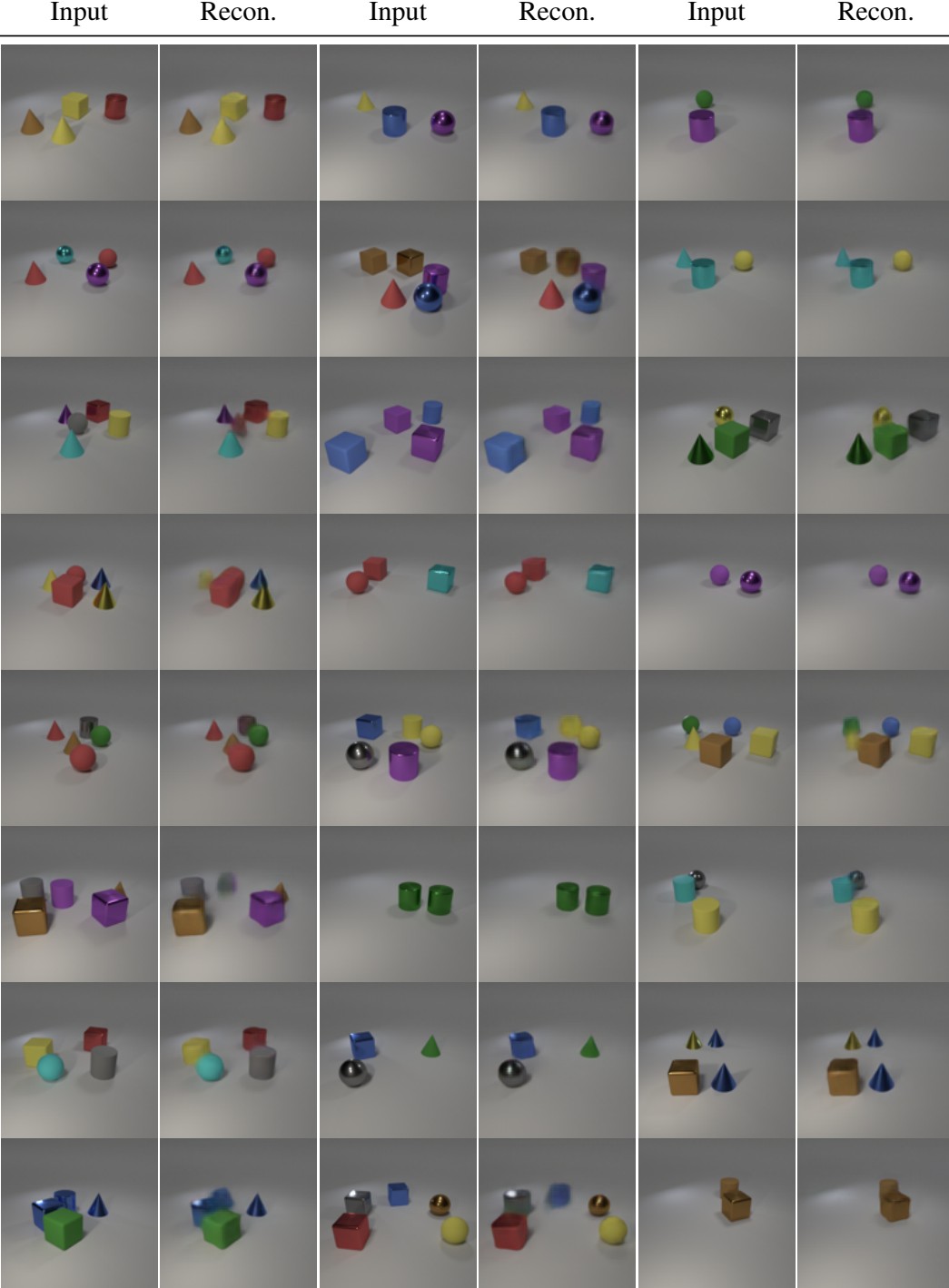

Figure 4: **Additional reconstructions on CLEVR2345.**

We have included additional manipulations of one scene in GIF format in the supplementary material. We show: i) each object rendered individually, ii) object identity swaps with other scenes, iii) object translations along one axis, iv) translations in diagonal (two axis), and v) objects moving in a circle.

| Input | Add | Remove | Input | Add | Remove |

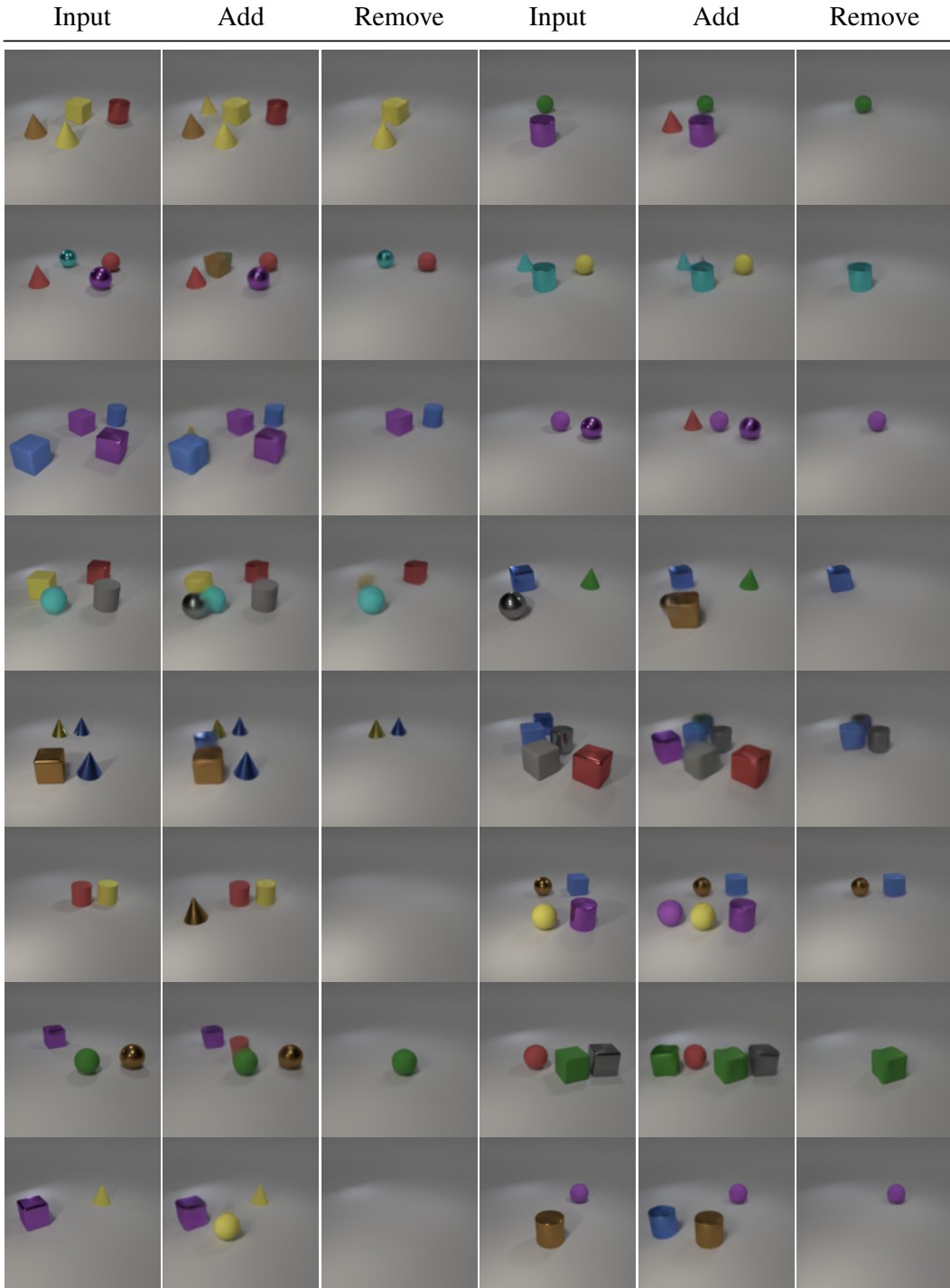

Figure 5: **Additional object additions and removals on CLEVR2345.** For each scene, we show images with one randomly added object, and with 1-3 random objects removed. Some of these images show out-of-distribution samples with a number of objects not seen during training (2-5 objects).

## D NOVEL VIEW SYNTHESIS

In this section we synthesize novel views of a scene by modifying the camera pose. For each example we show the input image, our reconstruction, then two images as a result of moving the camera $\pm 15°$ in the azimuth axis, and two images as a result of zooming in the scene. While our model is trained without ground-truth camera poses and with single views of scenes, it is able to generalize to small camera pose modifications and render novel views of a scene

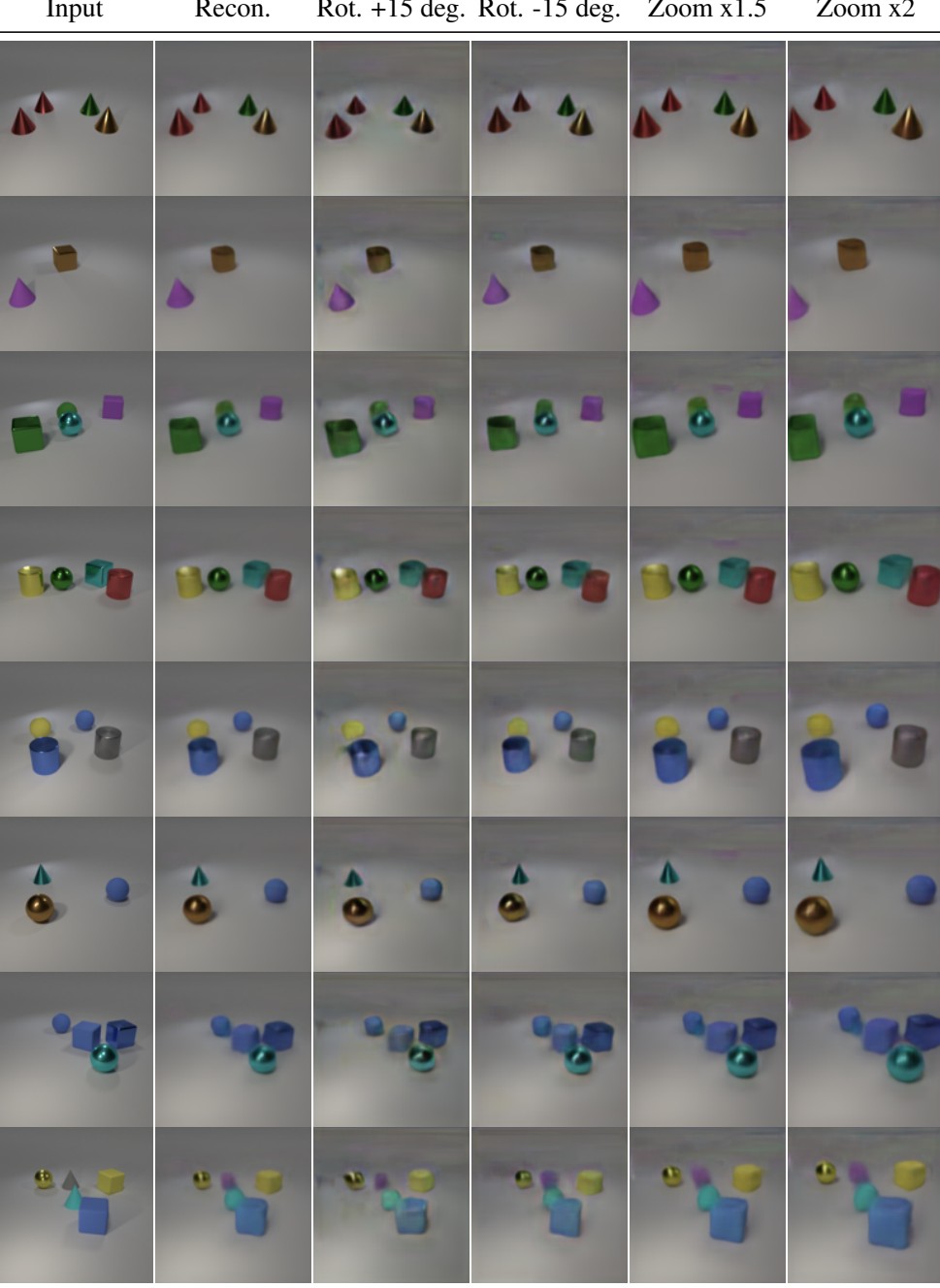

Figure 6: **Novel view synthesis on CLEVR2345.** For each scene, we move the camera $\pm 15°$ on the azimuth axis. Additionally, we zoom in the scene twice. While our model is trained with a fixed default camera pose and single scene views, it is able to generalize to small camera pose modifications and render novel views of a scene.

# E  NeRF OUTPUT

In this section we show the raw outputs of the NeRF function. These outputs show scene views rendered at low resolution, which are then upscaled with a neural network. While these generations have reduced details due to their resolution, they clearly show the different objects and their location in the scene. NeRFs are rendered at low resolution to ease the computational costs, as the time and memory requirements of rendering with a NeRF are linearly correlated with the number of casted rays/pixels in the output image.

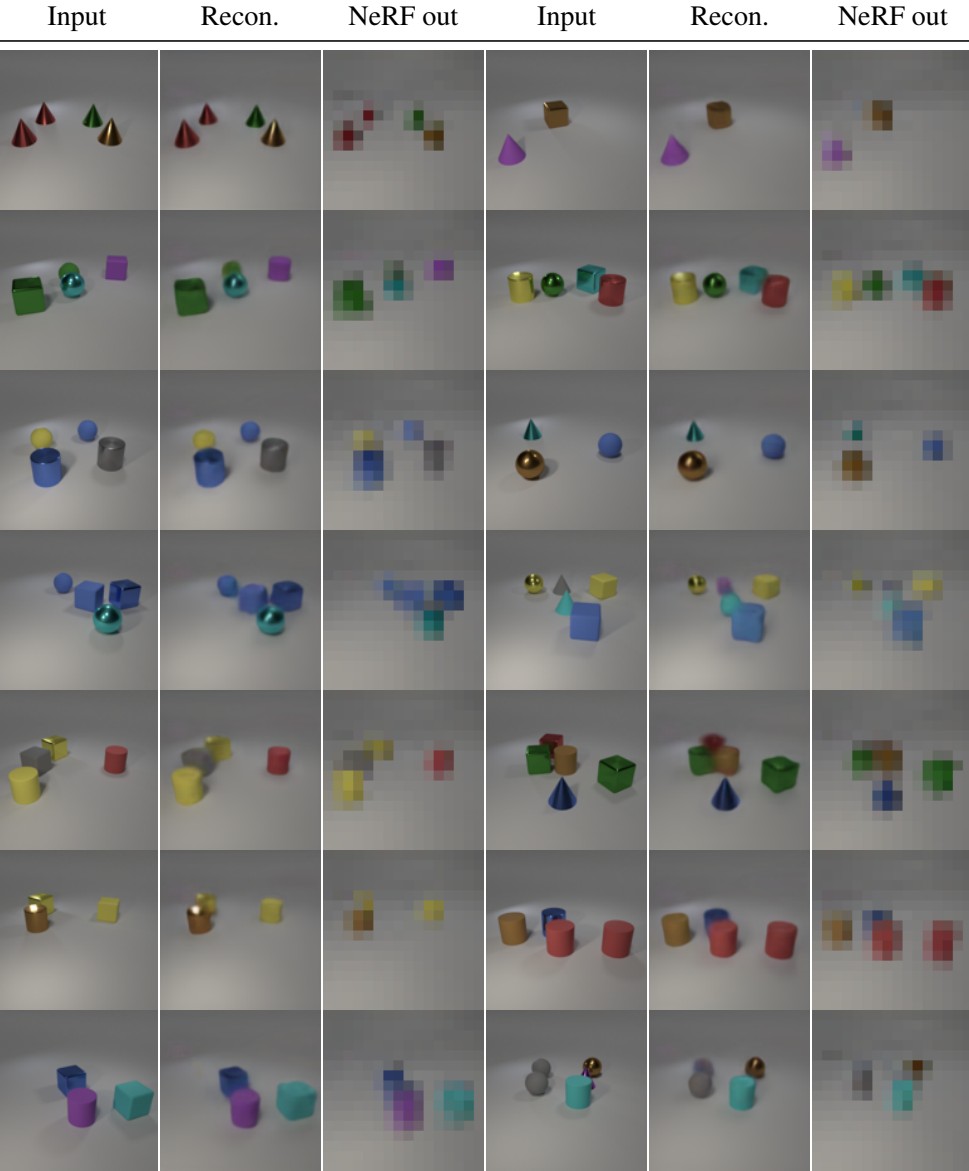

Figure 7: **NeRF outputs on CLEVR2345.** For each input scene we show the reconstructed image as well as each the low resolution output of the NeRF function. This output is then upscaled with a neural network to obtain the reconstructed scene. While NeRF output lack full detail, they correctly depict each individual object and their position in the scene.

# F  SHAPE - APPEARANCE DISENTANGLEMENT

In this section we show examples of the disentanglement of shape and appearance of objects. To demonstrate this property of our representations, we first reconstruct a given scene, and we then randomly change the appearance vector of one of the objects in the scene by the appearance vectors of other objects in the same scene. While the shading and color of the object changes, the overall shape remains the same.

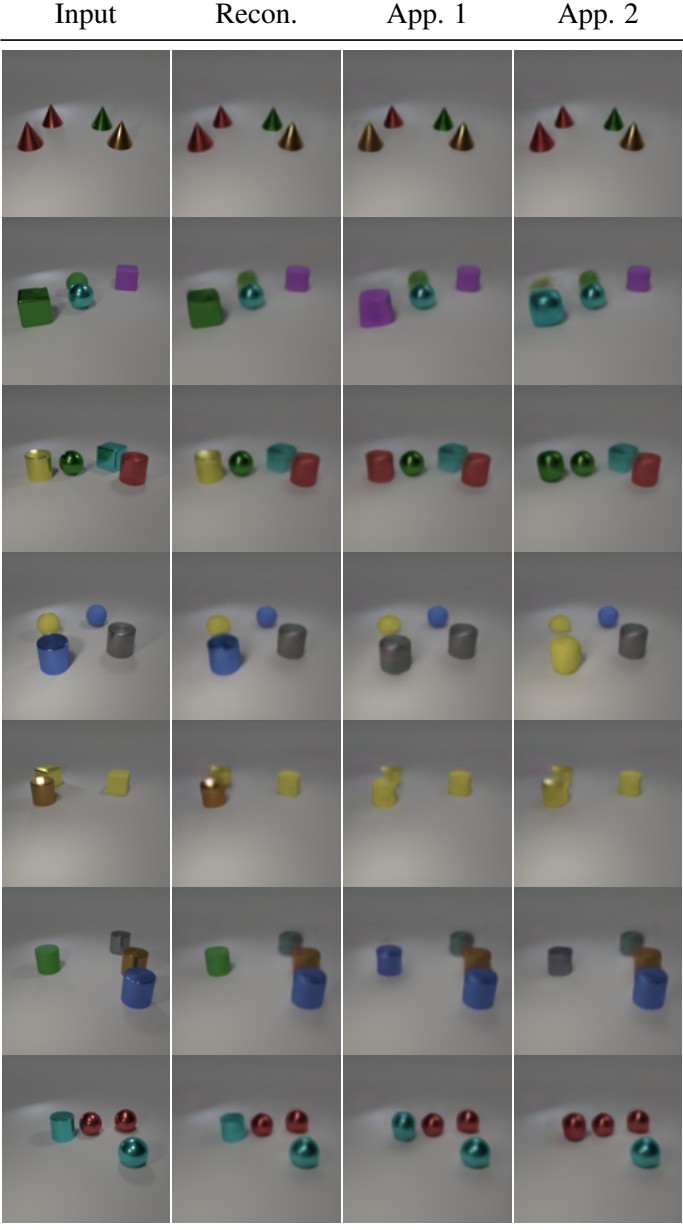

Figure 8: **Shape and appearance disentanglement on CLEVR2345.** For each input scene we alter the appearance vector of an object in the scene. First, we show the reconstructed scene. Then, for one of the objects in the scene, we replace its appearance vector by that of another object in the scene, while keeping its shape vector fixed. We perform this operation two times. We observe that, while the shading and color of the object changes as we change the appearance vector, its shape remains constant.

# G  OCCUPANCY MAPS

In this section we show the occupancy maps for each rendered object. The occupancy maps are obtained by integrating rays going from the image plane to the scene according to the NeRF density, but without taking into account the RGB output. We observe that each object is rendering a part of the scene corresponding to a single object instance. NeRF outputs are rendered at low resolution and then upscaled to full resolution with a neural network. To perform unsupervised object discovery we use the outputs of slot attention, since they do not require any upsampling and thus can provide more accurate segmentations.

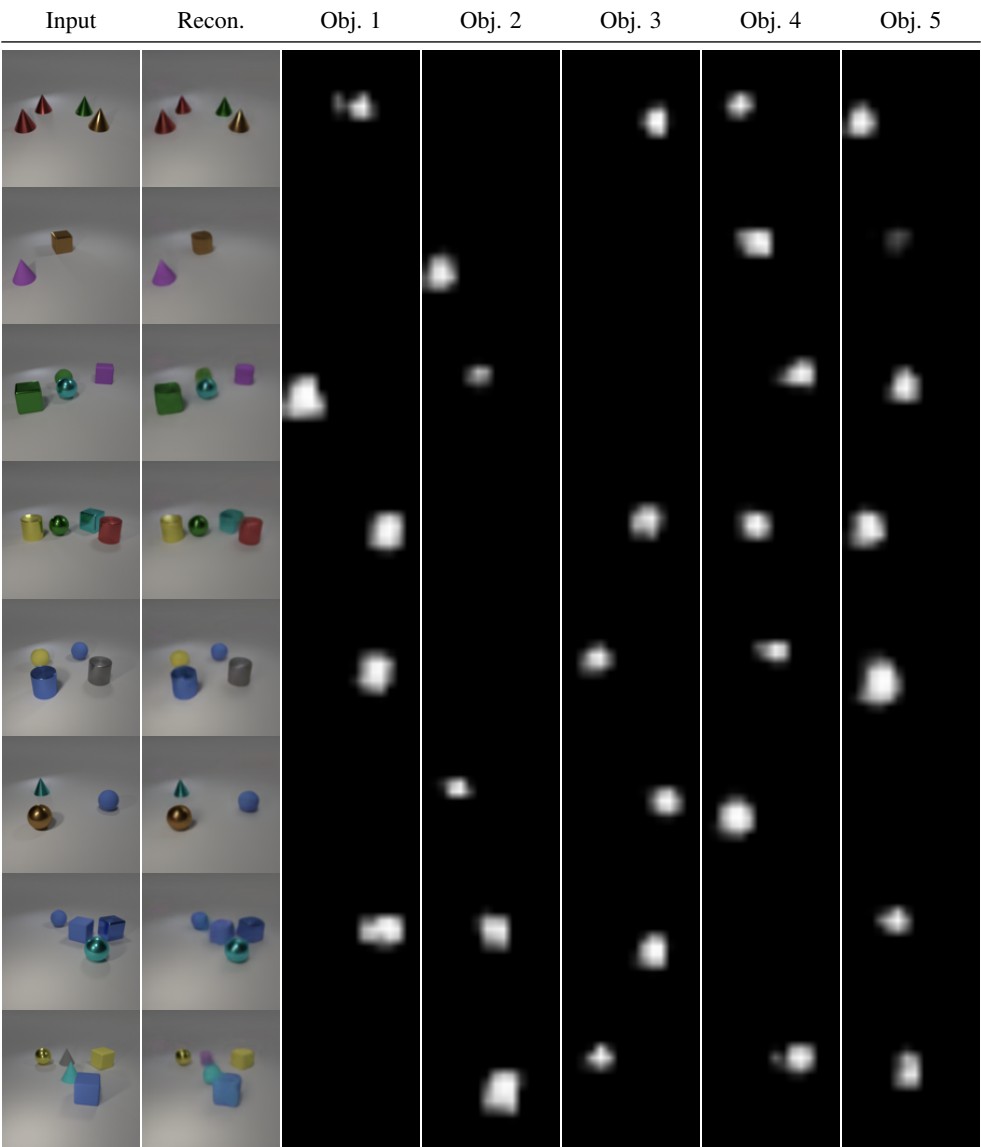

Figure 9: **Object occupancy maps on CLEVR2345.** For each input scene we show the reconstructed image as well as each individual object occupancy map. Occupancy maps are obtained by integrating the density outputs of the NeRF function along rays going from the image plane to the scene. We observe that each object is rendering a part of the scene corresponding to a single object instance. Outputs of the NeRF function have low resolution and are upscaled by a neural network. The last row shows an example of a bad grouping of two objects in the scene.

