# OpenReview forum: "INFERNO: Inferring Object-Centric 3D Scene Representations without Supervision"
_ICLR.cc/2022/Conference — ICLR 2022 Submitted_

### Official Review · Reviewer_3rCh · 2021-10-31

**Correctness:** 3
**Technical Novelty And Significance:** 2
**Empirical Novelty And Significance:** Not applicable
**Recommendation:** 5
**Confidence:** 4

**Details Of Ethics Concerns:**

N.A.

**Main Review:**

Strengths:

1. It is interesting to use NeRFs to represent objects in a scene, which encourages the understanding of 2D visual scene taking account into its 3D structure.
2. Explicitly modeling the 3D structure of 2D scene may help better deal with occlusion, ambiguity from similar appearance which can be a big challenge in 2D scene understanding.
3. It looks great to show how learned object features can be used for downstream tasks.

Weaknesses:
1. The technical novelty is somewhat limited. The paper mainly combines the Slot-attention and NeRF. Personally, it is hard to identify the exciting point(s).

2. The motivation of this work seems not well justified. It is claimed that the model is proposed to infer 3D representation for objects that are interpretable and amenable. But the experiments presented with such 3D representation cannot show its superiority and it is
hard to see the necessity of 3D object representation in this case considering the task presented can also be fulfilled with 2D object representation. In terms of the interpretability and manipulability of latent code, this paper presents objects removal/addition, object scaling and object translation. All of the above operations can actually be done just by explicitly modeling 2D object pose such as AIR (NIPS'16 paper). In terms of scene understanding in the form of object segmentation, this model cannot beat Slot Attention. In terms of downstream tasks, this model is not compared with other object-centric models. Thus, based on your current experiments, the motivation for learning such 3D representation for objects is not well justified.

3. Is it appropriate to use NeRF? If I understand correctly, NeRF learns the radiance field of a scene with 2D images rendered from different viewpoints. This model tries to recover the object radiance field from a single view of the objects. But the model training
objective can actually be fulfilled by just learning a single-view of the scene since there is only one image for each scene to generate reconstruction loss.

4. From my understanding, the 3D rendering model in this paper seems more similar to a decoder in VAE-wise architecture. It could be more convincing to provide some novel view synthesis (not novel scene) to claim it as a NeRF. For example, you may consider
object rotation.

5. All experiment datasets are a bit too simple. It is unclear how good/bad the proposed model will perform on realistic 3D scenes such as LLFF dataset or ScanNet. Otherwise, it is hard to see the real impact of this proposed method.


**Summary Of The Paper:**

This paper proposes a model to infer structured 3D object representations from a 2D scene in an unsupervised fashion so as to represent the visual scene in an object-centric way. Specifically, the inference part adopts a similar mechanism as Slot Attention to derive object
slot latent code, and then maps slot latent code to 3D object representations with MLP. The rendering part takes the idea from 3D neural rendering where a shared NeRF function is used to represent all objects excluding the background. Rendering is performed by querying NeRF with 3D location, 2D view direction as well as object latent to get object color value and occupancy value. Object rendering is composed into scene rendering according to location derived at the inference stage and weighted by occupancy value.

To sum up, the paper interprets a 2D visual scene with 3D object-centric representation. With the existing 2D object-centric scene segmentation method and 3D neural rendering approach, it achieves comparable segmentation performance and derives manipulable object representation.

**Summary Of The Review:**

Look forward to the response from authors.

---

> ### Author Response · Authors · 2021-11-22
> **Reply to reviewer**
>
> We thank the reviewer for their feedback. We answer their concerns below:
>
> * **Novelty of the approach**
>
> We are not aware of previous models that regress the 3D pose of objects in multi-object scenes without supervision and from a single view. We believe novelty is not fully measured by whether a model uses established components -  research builds upon previous discoveries - and instead the novelty of our method should be assessed on the basis that it tackles a different and relevant problem than previously proposed approaches.
>
> * **Motivation of the approach**
>
> Our model captures 3D scene information and can be used to perform edits of the 3D pose of objects. As such, we require a 3D representation of the scene, for which NeRFs are currently one of the best approaches, regardless of how many views are used to infer this representation. We do not believe that the edits we are performing on the scenes could be performed with 2D models. In particular, we are not aware of 2D models that can rotate the camera around the scene or move objects in a circle in a 2D plane in the 3D scene that is not axis-aligned with the image plane. We have added a 2D slot-attention baseline that shows that the representation learned by our model is superior for the CATER task than that of an equivalent 2D model.
>
> * **Why use NeRFs**
>
> We use NeRFs as one the best approaches to rendering 3D scenes.
>
>
> * **Our model being closer to a VAE-decoder than a NeRF**
>
> Our model is not a variational autoencoder as we do not perform variational inference and we do not model latent probabilities. Instead, our model is closer to a traditional autoencoder, with a differentiable 3D renderer. There is a distinct difference between using latent variable models and their training objective (VAE) and the approach used to render representations into 2D views (decoder or renderer, in our case NeRF). We do not simply "claim" that our model is a NeRF, but instead we *implement* rendering with neural radiance fields. In particular, we learn an implicit function that models RGB colors and density for 3D scene locations and 2D viewing directions.  We generate scene views by casting rays to the scene, evaluating the implicit function at different points along the casted rays and integrating the results according to the output density.
>
> * **Experiments are simple**
>
> We are not aware of any methods that can infer 3D object poses in multi-object scenes without additional supervision and from a single view. As such, we believe the datasets we use properly showcase the capabilities of our model and are adequate to propose initial research on this area. We are not aware of published unsupervised object-centric scene models (whether 2D or 3D) with an inference mechanism that are showcased on significantly more complex datasets than the ones we use.

---

### Official Review · Reviewer_G78k · 2021-11-02

**Correctness:** 3
**Technical Novelty And Significance:** 2
**Empirical Novelty And Significance:** 3
**Recommendation:** 5
**Confidence:** 4

**Main Review:**

Strengths:

* an interesting combination of slot attention and NeRF mixture models.
* paper is nicely presented
* results are nice, though see below

Weaknesses:

While I like the paper in general I feel there are several points missing which are probably worth addressing to:

* The objects are transformed using the pose - but that means the appearance is transformed as well - this could be a problem if some of the appearance is pose dependent, for example shading will change, or reflections. Can the authors comment how this is handled here? my guess is that the post rendering network (the upscaling one) takes some of this burden but that's probably not ideal in terms of representation.

* How does the model perform when scenes are rendered from other viewpoints? if the resulting representation is indeed 3D it should be no problem to render the scene from other viewpoints - is that the case here? if not, why? (again, my guess is the upscaling network may be problematic here as well)

* I see that the masks used for estimating segmentation performance (Figure 3) are from the slot attention mechanism - what happens if you use the masks as decoded from the slots (i.e. render the densitiy of each slot separately). This would be more convincing to demonstrate these are "3D" segmentation.

* I feel the experimental validation is quite limited - using only a single dataset (even if it's two variants) is not very convincing as to the generality of the method. How does it work on other datasets? more/less complex.

* I would be happy to see more discussion of the post rendering / upscaling network - this seems like a cruicial part and is not discussed enough.

**Summary Of The Paper:**

This paper proposes a model which is able to segment 3D scenes into objects by a combination of slot-attention (For inference) and a mixture of object NeRF functions which mix together (in 3D) to compose a scene. The method receives a single input image (with the camera coordinates though these are fixed) and extracts a set of slots - one slot for each object. These slots are decoded using a NeRF renderer: one part (the shape) generates the density, one (the appearance) generates the colours and one (the pose) transforms the points of the object to the appropriate pose in scene space. Results are demonstrated on CLEVR data as well as CATER (which is visually very similar) and some downstream tasks.

**Summary Of The Review:**

All in all a nice paper but I think it suffers from some drawback as discussed above.

---

> ### Author Response · Authors · 2021-11-22
> **Reply to reviewer**
>
> We thank the reviewer for their feedback. We answer their concerns below:
>
> * **Appearance across camera locations**
>
> We agree that the appearance might change at different poses, that is why when we infer it it is a function of the pose. Note that in our manipulating scene experiments, we change the pose vector but reuse the same appearance vector inferred as it gave a good result for this experiment. We have added some visualizations of the low resolution NeRF output to show the effects of the upscaling network.
>
> * **Novel view synthesis**
>
> We have included some examples of camera zooming and rotation. Our model can generate novel views but is limited in the range of camera transformations it can apply, since we do not use ground-truth camera positions and we use a single view of each scene with the same fixed camera position. This makes it hard to infer the appearance of the invisible parts of the objects. However we are not aware of any model that infers better 3D geometry without any supervision and from a single scene view.
>
> * **Masks from rendered objects instead of slots**
>
> We have included some examples of the object masks inferred by using the output density of the NeRFs. They correspond to the object masks. However, since the output of the NeRF is at low resolution, we would have to upscale them to compute the ARI metric for unsupervised object discovery, and that is why we decided to use the slot masks instead, as they are at full resolution.
>
> * **Experiments on other datasets**
>
> The main limitation for modeling more complex datasets is to have a valid unsupervised scene decomposition. Slot Attention has difficulties segmenting harder scenes than those in CATER. At the same time, we would like to note that we are not aware of object-centric 3D methods that operate on harder datasets. Slot Attention, G-SWM and other methods that perform object-centric scene inference without supervision are demonstrated on CLEVR or simpler datasets.
>
> * **Upscaling network**
>
> Using an upscaling network for upscaling NeRF outputs is not uncommon in the literature (e.g. GIRAFFE). The motivation to use it is to reduce computational requirements, since the time and memory complexity of rendering with NeRFs is a direct function of the number of rays casted, which depends on the output resolution. In the future we plan to remove the network by making our rendering process more efficient, as the sole purpose of the network is to upscale images, and in doing so it can produce issues due to not generalizing to unseen inputs (for example when upsampling scenes that are outside the training distribution). We have included examples of the low resolution NeRF output to validate that the upscaling network is just learning to upsample the low-resolution output to the proper resolution without changing its content.

---

> > ### Comment · Reviewer_G78k · 2021-11-25
> > **Thank you for the detailed response and revised paper.**
> >
> > Regarding the upscaling network - I would say I am now more worried than I was before :) it seems that the output from the NeRF model is VERY low resolution and a lot of the work falls on the upscaling network. This would explain the high quality reconstructions but also explains cases like Fig 8 - where often the shape and appearance seem quite mixed (for example bottom row, where replacing the appearance also changes the shape from cylinder to sphere). This is also reflected in the mask output from from the NeRF model (The occupancy visualisation). These are very very low res - they may allow for some object localisation but not much beyond.
> >
> > Regarding the novel view synthesis - yes, it is not easy to do this from a single view (at least in training, it is possible to train on multiple views and then infer from a single one as in NeRF-VAE). Nevertheless these results are reasonable. I actually suspect the upscaling network hurts here as it is predominantly 2D and would struggle to cope with some more complex 3D transformations.
> >
> > Having thought more about the pose estimation - as other reviewers have mentioned - it is not strictly accurate to say that the pose is inferred here in the sense that the coordinate system of the pose is arbitrary (and can always be compensated by an inverse shift of the representation). At most you could do relative operations here - but nothing guarantees real 3D pose estimate. With multiple views this would be possible I think.
> >
> > In any case I have decided to keep my score as is - I think it's a decent paper which can be even better.

---

> > > ### Author Response · Authors · 2021-11-28
> > > **Reply**
> > >
> > > Thanks again for your feedback and for engaging in the discussion. We reply to your comment below:
> > >
> > > Regarding the low resolution output of the model, this is a conscious design decision. Each time we double the image side, the amount of rays that have to be computed increases 4x, and so do the memory and computation requirements. This makes training at such resolutions expensive (for 128 pixel images, it would involve a x64 increase in memory) and we believe that having a hybrid radiance field + NN upscaling rendering pipeline to ease the computational costs is a valid solution, as long as it does not change the composition of the scene (which we have never observed). As we mentioned, this is also not something novel about our work, but instead an accepted solution to the computational cost - fidelity dilemma.
> > >
> > > In the example mentioned for the disentanglement of shape and appearance, the shape of the object actually does not change - only the appearance changes, which for this rendering model also includes *shading*. Therefore, the cylinder looks metallic like the other balls and loses the shading of the edge between the body and the top face of the cylinder, but you can clearly see that the shape is not that of a ball.
> > >
> > > Regarding the pose coordinate system - We do model a camera extrinsics matrix that maps from camera space to world space, so any poses inferred relative to a camera can be mapped to world space. We are using a fixed camera and the mapping to world space is not ground truth. However, multiple views do not solve the problem. Without a ground truth mapping from one camera location to world space (and that would mean additional supervision), inferred poses can only be relative to the camera, and their mapping to world space coordinates is still going to be unspecified. That is because, without a reference point from the camera space to world space, we do not know where the world coordinate system origin is nor the direction of their axis. Any rotation or translation of the world space origin defines a different mapping from camera space to world space, and even if the scene is rendered consistently across views, this mapping cannot be established without ground truth information.

---

### Official Review · Reviewer_k83T · 2021-11-02

**Correctness:** 3
**Technical Novelty And Significance:** 3
**Empirical Novelty And Significance:** 3
**Recommendation:** 5
**Confidence:** 4

**Main Review:**

The paper tackles a very challenging problem, is well-written and easy to follow, and has a good set of experiments for testing object-centric representations. However, my main concern is that the experiments did not focus on testing the '3D-ness' of the inferred representations.
- For scene segmentation, reconstruction, and manipulation, each scene is tested only on a single camera viewpoint. It is unclear whether the learned representations can capture the object appearance from viewpoints different from the input view. Previous work [1, 2] demonstrates this through novel view synthesis.
- The results on the CATER snitch localization task did not strongly support that the '3D' representations learned by the proposed model are more beneficial than 2D representations. One reason might be that this task relies more on long-term reasoning than on 3D reasoning. The authors may want to consider some other tasks that can clearly show the benefit of having 3D representations, like the ones used in [2]. Also, instead of comparing to Aloe, a more direct comparison would be to compare with Slot Attention combined with the same 12-layer transformer used for the proposed model.

**MINOR COMMENTS**
- The rendering function is denoted $g_\gamma$ in Equation 1 but $g_\tau$ in Section 2.1.
- From the paragraph directly below Figure 1, it seems the model can infer the camera location $c$. However, this was not demonstrated in any of the experiments.
- The object NeRFs are shared, which may only be able to deal with similar objects. The paper could be strengthened by experiments on more complex scenes, like the ones used in [1].
- What is the reference frame for the pose vectors? It is mentioned in Appendix B.1 that the manipulations are done by moving the objects forward on the Z or X axes. How are the XYZ axes defined? I would imagine that under different camera viewpoints, the same manipulation would have different effects, but the results in Figure 2 seem quite consistent.

[1] [Decomposing 3D Scenes into Objects via Unsupervised Volume Segmentation](https://arxiv.org/abs/2104.01148).

[2] [ROOTS: Object-Centric Representation and Rendering of 3D Scenes](https://arxiv.org/abs/2006.06130).

**Summary Of The Paper:**

The paper aims to decompose a scene into objects and infer the representations of 3D occupancy, color, and pose for each object from a single image of the scene without supervision. To this end, the paper proposes an autoencoding solution by combining the Slot Attention encoder with the GIRAFFE decoder. Each object is represented as a Neural Radiance Field (NeRF) additionally parameterized by the latent variables inferred from the encoder. The decoder then compositionally renders the objects. The experiments show that the proposed model (1) achieves competitive 2D segmentation performance on CLEVR6, (2) supports object-wise scene manipulation, and (3) outperforms non-object-centric methods on CATER snitch localization when combined with a powerful transformer.

**Summary Of The Review:**

I recommend reject, because the main claim, learning 3D representations, is not well supported by the experiments, and the benefit of the learned representations over prior work is not well demonstrated.

---

> ### Author Response · Authors · 2021-11-22
> **Reply to reviewer**
>
> We thank the reviewer for their feedback. We answer their concerns below:
>
> * **Novel View Synthesis**
>
> We have added an experiment in the appendix where we show novel views of the scene by zooming in the camera or panning the camera. We note that the capabilities of our model to generate novel views are limited in the range of transformations compared to the references indicated. This is because we only use a single view for each scene and a fixed camera pose throughout the dataset. ObjSURF and ROOTS use i) ground-truth camera poses and ii) multiple views of the same scene, which is significant additional supervision. Additionally, ObjSURF uses RGBD inputs with depth information. Nevertheless, we demonstrate in this new experiment that our model is able to generate novel views of a 3D scene and thus infer a 3D scene representation.
>
> * **Comparison to 2D baseline on CATER**
>
> We have included a 2D slot-attention baseline on CATER, using the same 12-layer transformer as in our proposed model. Our model significantly outperforms it, indicating that there is an advantage to using 3D representations instead of 2D representations. There are baselines that outperform our model, however they are explicitly designed for the CATER task as opposed to our model.
>
> * **Reference frame for the pose vectors**
>
> The pose vectors are regressed in world space, and transformations are applied using the world coordinate system. Since we are not using ground-truth information on the camera pose, we are using a fixed camera looking at the XY plane with a certain altitude, and therefore the transformations are consistent across scenes. We have added a figure where we show that our model can extrapolate to modifications of the camera pose without seeing different cameras during training. Under these modified camera positions the transformations are still consistent with the world coordinates, but are no longer aligned with the camera axes.

---

> > ### Comment · Reviewer_k83T · 2021-11-28
> > **Thanks for your detailed response**
> >
> > The added experiments do improve the paper, and I decided to raise my score to 5.
> >
> > I still have the concern that most quantitative evaluation (e.g., segmentation and reconstruction) is still done on a single camera viewpoint. I think more quantitative evidence is needed to support the claim that the model learns 3D representation. For example, you can feed a novel camera viewpoint to the model at test time, and calculate the segmentation and reconstruction metrics for that viewpoint. Also, you may measure the error of the inferred object pose with respect to groundtruth.

---

### Official Review · Reviewer_noTD · 2021-11-05

**Correctness:** 2
**Technical Novelty And Significance:** 3
**Empirical Novelty And Significance:** Not applicable
**Recommendation:** 5
**Confidence:** 5

**Main Review:**

Strengths:

+ The paper is clearly written and easy to follow.
+ It is novel and interesting to see object-centric learning models to infer 3D poses.
+ Validating the usefulness in downstream visual reasoning task is a plus.

Weakness:

My main conern is in the insufficient experiments that do not support the main contributions, i.e., the experiments do not demonstrate the benefits of the proposed 3D, structured object representations.

As for representations being structured (i.e., disentangled shape, appearance and poses):
- For disentanglement of shape and appearance, there is no experiments demonstrating this point. For example, this can be easily shown by swapping appearances of two objects in the scene while keeping their shapes unchanged, and vice versa.
- For disentanglement of 3D pose, the experiments include scaling and translation. However, there is no comparison to 2D baseline methods, which are able to do scaling and translation (at least the experiments on moving to the "right").

As for representations being 3D, there is no comparisons to show the benefits of 3D representation over 2D. In segmentation experiments, the results are worse than 2D baselines. In the visual reasoning task, the performance is worse than Aloe. Perhaps Aloe is better because it is designed for that task, but there is neither comparisons to 2D general object-centric baselines (such as original slot attention) or unstructured baselines.

Other minor weaknesses include:
- No ablation study on the model components to show the benefits of proposed technical contributions, such as the effect of regressing pose vs. not regressing pose (I suppose this should affect background decomposition because that is the only inductive bias that separates background slot from foreground slots).
- It would be good to demonstrate on datasets other than CLEVR.

**Summary Of The Paper:**

This paper proposes a novel unsupervised scene decomposition model that infers object shapes, appearances and 3D poses. The benefits over existing models are the structured, 3D object representations which allows to manipulate objects in the scenes such as moving and replacing objects. This paper also shows that the inferred object representations can be used in a visual reasoning task.

**Summary Of The Review:**

Although this paper is interesting in the ability to infer 3D properties of objects, the fundamental flaws in experiments are the main reasons that I recommend revision and resubmission.

---

> ### Author Response · Authors · 2021-11-22
> **Reply to reviewer**
>
> We thank the reviewer for their feedback. We answer their concerns below:
>
> * **Disentanglement of shape and appearance**
>
> To disentangle the shape and appearance of the objects we use the formulation in GRAF (Schwarz et al. 2020). GRAF makes the RGB output (appearance) be a function of the appearance latent, and the density output (shape) be a function of the shape latent. Therefore, each latent cannot affect the other property by design. We have included an example of a scene where we change the appearance of the object in the supplementary material while keeping the shape fixed. This is not a novel contribution of our paper and when describing this part of the model in Section 3 we cite and indicate that this was already done in GRAF.
>
> * **Disentanglement of pose**
>
> We show that we can move objects in a circle in a 2D plane inside the 3D scene in the zipped supplementary material. We have additionally included examples where we move the camera in the 3D scene in the document appendix and supplementary material. While some 3D transformations can be approximated by scaling and translations on the image plane, transformations we showcase such as camera panning do not have a direct 2D equivalent. We thus empirically demonstrate that we can perform 3D pose manipulations.
>
> * **Advantages of a 3D model**
>
> One of the goals of the paper is to show that we can learn 3D structure from single 2D views of a scene by using a structured renderer, as well as learn to perform inference and train this model end-to-end. The main advantage of the model is in being able to perform 3D pose manipulations, which 2D methods cannot. In terms of the CATER experiment, we have included a slot attention 2D baseline. Our model outperforms this baseline significantly.
>
> * **No ablation of regressing or not the object pose**
>
> Performing inference of 3D object poses is one of the main contributions of our approach and a crucial part of our rendering process. It would not be possible to perform interpretable pose manipulations as we show without inferring the object poses.

---

> > ### Comment · Reviewer_noTD · 2021-11-29
> > **Thank you for your response**
> >
> > Thank you for your response! The added 2D baseline comparisons do improve the paper in terms of the application on visual reasoning task which is an important factor in my consideration, so I increased my score to 5. But I still have concerns in terms of the claimed disentanglement (and the advantages) not well demonstrated.
> >
> > For 2D methods such as original slot attention, it represents an object by a segment of image. So it should be able to do at least in the "moving to the right" experiment, by simply moving the image segment. So this is not undoable for 2D methods. I would guess it lacks normal shading effects and shows some artifacts, though, and I suggest this experiment might demonstrate the advantage of the proposed model being 3D-aware.
> >
> > I would encourage the authors to improve this.

---

### Author Response · Authors · 2021-11-22
**Summary of changes**

We would like to thank all reviewers for their valuable feedback. We have updated the document with the following changes:

* **Novel view synthesis**

We have added visualizations of the same scene from different camera locations and with different zoom levels. We have added examples in the appendix and have included a video where we interpolate camera positions in the supplementary material. We hope that these results further demonstrate that we learn 3D scene representations.

* **Slot Attention baseline**

We have added a Slot Attention baseline for the CATER task. This baseline utilizes a slot attention model pretrained on the CATER dataset. The mask decoder is then discarded and we use the same 12-layer transformer model that we use for INFERNO to predict the snitch location from the Slot Attention representation, which is fine-tuned for the task. This baseline obtains 59.1% Top-1 accuracy, significantly below the 71.1% Top-1 accuracy obtained by our model. We hope this convinces the reviewers that our 3D representations offer an advantage on the CATER task compared to 2D scene representation learning baselines.

* **Visualizations of the low resolution NeRF rendering**

We added visualizations of the low resolution NeRF rendering performed by our model. These visualizations validate that the upscaling neural network does not modify the scene structure.

* **Visualizations of the shape and appearance disentanglement**

We have added visualizations that reinforce that the object shape and appearance are disentangled in our model, by showing the same shape being rendered with different appearances.

* **Visualizations of the occupancy maps for each object**

We have added visualizations of the image regions represented by each object in our model. These maps are generated with the density outputs of our model and therefore are low resolution.

---

### Decision · Program_Chairs · 2022-01-20

**Decision:**

Reject

**Comment:**

The paper proposes an approach for learning a decomposition of a scene into 3D objects using single images without pose annotations as training data. The model is based on Slot Attention and NeRF. Results are demonstrated on CLEVR and its variants.

The reviewers point out that the method is reasonable and the paper is quite good, but even after considering the authors' feedback agree that the paper is not ready for acceptance. In particular, the key concern is around experimental evaluation - that it is performed on one dataset (and variants thereof) and that the evaluation of the 3D properties of the model is not sufficiently convincing: it does not outperform 2D object learning methods on segmentation and is not compared to those on "snitch localization".

Overall, this is a reasonable paper, and the results are promising but somewhat inconclusive, so I recommend rejection at this point, but encourage the authors to improve the paper and resubmit to a different venue.

(One remark. The paper makes a point of not using any annotation. It is technically true, but in practice on CLEVR unsupervised segmentation works so well that it's basically as if segmentation masks were provided. If the authors could demonstrate that their method - possibly with provided coarse segmentation masks - works on more complex datasets, it would be a nice additional experiment)